# Quantitative Changes in Vascular and Neural Fibers Induced by Subretinal Fluid Excluding the Peripapillary Region in Patients with Chronic Central Serous Chorioretinopathy

**DOI:** 10.3390/diagnostics15020174

**Published:** 2025-01-14

**Authors:** Esra Kızıldağ Özbay, Şenol Sabancı, Mehmet Fatih Küçük, Muhammet Kazım Erol

**Affiliations:** Department of Ophthalmology Antalya Training and Research Hospital, Varlık, Kazım Karabekir Cd., 07100 Antalya, Turkey; sbncsenol@yahoo.com (Ş.S.);

**Keywords:** central serous chorioretinopathy (CSCR), retinal nerve fiber layer (RNFL) thickness, radial peripapillary capillary (RPC) vessel density, subretinal fluid (SRF), optical coherence tomography angiography (OCTA)

## Abstract

**Background:** This study aims to evaluate the quantitative changes in retinal nerve fiber layer (RNFL) thickness and radial peripapillary capillary (RPC) vessel density in patients with chronic central serous chorioretinopathy (CSCR), specifically excluding the peripapillary region. **Methods:** A prospective case–control study was conducted at the Antalya Training and Research Hospital, Health Sciences University, involving 65 patients with chronic CSCR. Participants were categorized into two groups based on the presence or regression of subretinal fluid (SRF). A control group of age- and sex-matched healthy individuals was also included. Optical coherence tomography angiography (OCTA) was used to assess RNFL thickness and RPC vessel density. Statistical analyses were conducted using SPSS, with non-parametric tests employed for between-group comparisons. **Results:** Patients with persistent SRF exhibited significant increases in RNFL thickness in the inferior and nasal quadrants compared to healthy controls (*p* = 0.003 and *p* = 0.014, respectively). Additionally, RPC vessel density in the small vessel disc area (%) was significantly lower in the persistent SRF group compared to controls (*p* = 0.021). A significant negative correlation was found between nasal quadrant RNFL thickness and small vessel disc area (*p* = 0.014, r = −0.306). **Conclusions:** Chronic SRF in CSCR patients, even when not involving the peripapillary region, leads to significant structural changes in both the neural and vascular components of the retina. These findings suggest that SRF contributes to broader retinal alterations and supports the need for early detection and management of CSCR to prevent long-term visual impairment.

## 1. Introduction

Central serous chorioretinopathy (CSCR) is a retinal disorder commonly affecting middle-aged individuals and is characterized by the accumulation of subretinal fluid (SRF) due to dysfunction in the retinal pigment epithelium (RPE) and increased choroidal vascular permeability [1,2,3,4]. This leads to serous retinal detachment, which can cause visual symptoms such as blurred vision, metamorphopsia, and central scotomas. Prolonged subretinal fluid accumulation can damage photoreceptors, contributing to permanent vision loss. While the pathogenesis of CSCR is not fully understood, it is thought to involve impaired choroidal vascular regulation and increased choroidal vascular permeability [5,6,7].

Chronic central serous chorioretinopathy, typically defined in the literature as subretinal fluid persistence for more than 3 months, can lead to significant structural changes in the retina and choroid. These changes include thinning of the retinal nerve fiber layer, atrophy of the retinal pigment epithelium, and choroidal vascular remodeling. However, the quantitative changes in vascular and neural fibers in the subretinal fluid-affected region, excluding the peripapillary area, have not been extensively studied [6,8,9,10,11].

CSCR presents in two main clinical forms: acute and chronic. While acute CSCR often resolves spontaneously, chronic CSCR can lead to persistent visual symptoms and more profound structural changes in the retina and choroid. In this study, the definition of persistent SRF was adapted to include cases where subretinal fluid remained detectable for more than six months, aligning with stricter inclusion criteria aimed at identifying stable and chronic cases. By doing so, we aim to evaluate the structural changes associated with chronic CSCR in patients without subretinal fluid in the peripapillary region, specifically focusing on retinal nerve fiber layer (RNFL) thickness and radial peripapillary capillary (RPC) vessel density.

## 2. Materials and Methods

This prospective case–control study was conducted at the Ophthalmology Clinic of Antalya Training and Research Hospital, affiliated with the Health Sciences University. The study focused on patients diagnosed with chronic central serous chorioretinopathy (CSCR) who had been under consistent follow-up for a minimum period of one year. Patients were carefully divided into two distinct groups based on the status of their sub-retinal fluid (SRF): one group consisted of individuals with persistent SRF, while the other included those with regressed SRF. In this study, persistent SRF was defined as subretinal fluid that remained detectable for more than six months, consistent with definitions commonly used in the literature. To provide a baseline for comparison, an age- and sex-matched control group of healthy individuals with no history of CSCR was also incorporated into the study.

The inclusion criteria for the study were as follows: participants were required to have a confirmed diagnosis of chronic CSCR for at least six months, no signs of subretinal fluid in the peripapillary region for at least one year, and be matched by age and sex to the individuals in the control group. Conversely, the exclusion criteria involved any presence of subretinal fluid in the peripapillary region at any point during follow-up, recurrence of CSCR within the study period, and any history of intraocular surgery or intraocular inflammatory conditions, as these factors could confound the study results.

Each participant underwent thorough visual acuity and intraocular pressure evaluations. Visual acuity was assessed using the logMAR scale, ensuring precise measurement of visual function, while intraocular pressure was measured using standard tonometry methods. These assessments allowed for a comprehensive analysis of any visual and pressure-related variations between the CSCR patient groups and the control group, contributing to a deeper understanding of the potential impact of SRF status on visual outcomes in chronic CSCR cases.

### 2.1. Assessment of Retinal Nerve Fiber Layer Thickness and Radial Peripapillary Capillary Density

In this study, Optical Coherence Tomography Angiography (OCTA) was employed as a key imaging modality to assess retinal nerve fiber layer (RNFL) thickness and radial peripapillary capillary (RPC) vessel density in all participants. RNFL thickness was meticulously measured across five distinct regions, which included the peripapillary region as well as the superior, inferior, temporal, and nasal quadrants. These regions were chosen to allow for a comprehensive understanding of RNFL thickness variations across different parts of the retina.

Additionally, RPC vessel density was evaluated by examining several specific areas that are critical to understanding vascular health and distribution within the retina. Measurements were taken in the small vessel total area, small vessel disc area, and small vessel peripapillary area, as well as in the total vessel total area, total vessel disc area, and total vessel peripapillary area. This multi-regional analysis of both RNFL thickness and RPC vessel density provides valuable insights into the microvascular and structural characteristics of the retina, particularly in the context of chronic CSCR. By employing OCTA in this detailed manner, the study aims to capture subtle yet clinically significant variations in retinal structure and blood flow, contributing to a deeper understanding of the underlying pathophysiology in affected individuals.

### 2.2. Statistical Analyses

The statistical analysis of the data was carried out using SPSS 26.0 program SPSS software, a widely recognized tool for performing complex data analyses in clinical research. To determine whether the data followed a normal distribution, the Shapiro–Wilk test was applied as the initial step. For data that did not meet the assumptions of normality, the non-parametric Mann–Whitney U test was used to assess differences between two independent groups, providing a robust approach for analyzing skewed data.

For comparisons involving multiple groups, the Kruskal–Wallis test, a non-parametric equivalent of ANOVA, was employed to detect any statistically significant differences. When significant differences were identified among the groups, post hoc analyses were conducted using the Bonferroni correction to control for Type I errors, thereby ensuring more accurate identification of specific group differences.

Additionally, correlations between variables were explored using Spearman’s correlation coefficient, which is particularly suitable for non-normally distributed data. In this study, a *p*-value threshold of less than 0.05 was set to determine statistical significance, allowing for careful interpretation of the results. This comprehensive approach to statistical analysis was chosen to ensure that all findings are both rigorous and reliable, lending greater confidence to the conclusions drawn from the study data.

## 3. Results

### 3.1. Demographic and Clinical Characteristics of Participants

A total of 65 patients with chronic CSCR participated in the study, of which 38 had persistent SRF and 27 had regressed SRF. The groups were comparable in terms of age and sex (*p* = 0.457). Visual acuity was significantly lower in the group with persistent SRF compared to the group with regressed SRF (*p* < 0.001). No significant differences were found in intraocular pressure between the two groups (*p* = 0.469) (Table 1).

### 3.2. Radial Peripapillary Capillary Vessel Density (RPC VD)

In the persistent SRF group, RPC vessel density in the small vessel disc area (%) was significantly lower compared to the healthy control group (*p* = 0.021). No significant differences were observed in other vessel density measurements, such as total vessel area or disc area. Table 2 demonstrates the comparison of RNFL thickness and RPC VDs between chronic CSCR patients with and without SRF and controls. The extensive fluid pocket observed in the peripapillary area and its impact on retinal architecture are illustrated in Figure 1, which highlights the structural distortion caused by subretinal fluid.

Subretinal fluid: The visualization demonstrates an extensive fluid pocket, which extends into the peripapillary area, directly impacting the optic nerve head.Structural changes: The image highlights how the presence of fluid distorts the normal retinal architecture, including potential thinning of the surrounding neural layers and vascular structures.Clinical relevance: This degree of peripapillary involvement emphasizes the importance of early detection and management in CSCR to prevent long-term complications such as optic nerve compression or irreversible damage.

**Figure 1 diagnostics-15-00174-f001:**
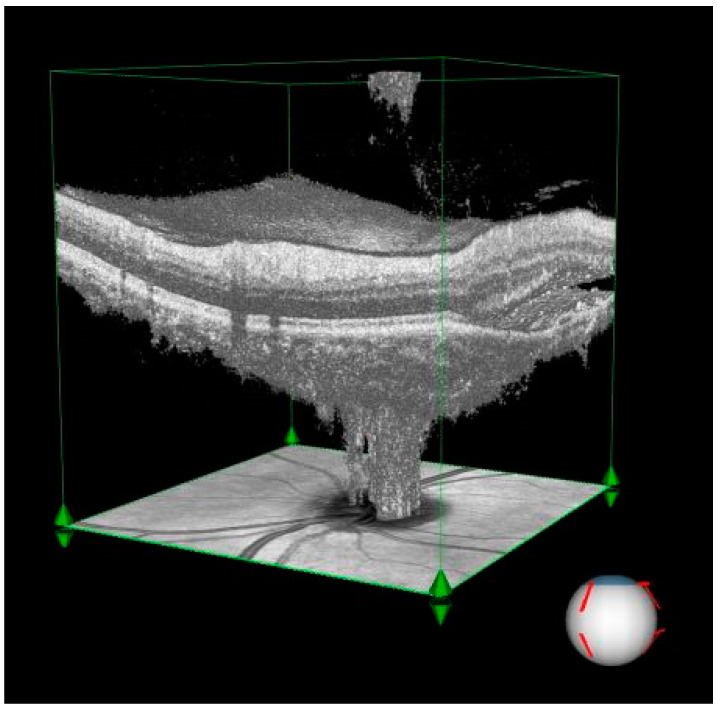
Three-dimensional visualization of optical coherence tomography (OCT) imaging illustrating significant subretinal fluid accumulation in the peripapillary region of a patient with central serous chorioretinopathy (CSCR).

**Table 2 diagnostics-15-00174-t002:** Comparison of RNFL thickness and RPC VDs between chronic CSCR patients with persistent SRF, regressed SRF, and controls.

		CSCR Patients with Regressed SRF Median (Min–Max) *n*: 27	CSCR Patients with SRF Median (Min–Max) *n*: 38	Control Median (Min–Max) *n*: 65	*p* Value
RNFL (µ)	Peripapillary	112.00 (97.00–145.00)	117.00 (92.00–151.00)	113.00 (98.00–140.00)	0.242
	Superior	136.00 (114.00–239.00)	132.00 (101.00–175.00)	130.00 (105.00–165.00)	0.489
	Inferior	50.20 (48.00–72.00)	51.90 (45.00–88.00)	50.30 (47.00–55.00)	0.003
	Temporal	148.00 (119.00–206.00)	146.00 (123.00–196.00)	142.00 (114.00–190.00)	0.609
	Nasal	74.00 (50.00–89.00)	78.00 (60.00–115.00)	75.00 (58.00–113.00)	0.014
RPC VDs (%)	Small Vessel Whole Area	106.00 (48.30–162.00)	102.00 (46.80–164.00)	100.00 (71.00–152.00)	0.599
	Small Vessel Inside Disc Area	50.40 (36.70–57.10)	49.50 (41.10–57.70)	52.00 (37.50–59.20)	0.021
	Small Vessel Peripapillary Area	53.10 (48.40–58.60)	52.55 (45.30–57.50)	52.90 (49.40–57.80)	0.331
	All Vessel Whole Area	56.90 (54.10–61.00)	57.05 (51.50–60.90)	57.30 (52.70–61.00)	0.987
	All Vessel Inside Disc Area	61.30 (50.40–67.40)	59.70 (47.90–66.80)	61.40 (47.10–67.90)	0.090
	All Vessel Peripapillary Area	59.80 (55.80–64.10)	59.25 (45.80–63.50)	59.20 (56.20–63.80)	0.439

The three-dimensional perspective enhances the understanding of how chronic CSCR affects both retinal and optic nerve structures, making it a valuable diagnostic and educational tool.

Table 2 provides the median (min–max) values and a single *p*-value from the Kruskal–Wallis test comparing the three groups. For those parameters with a significant overall *p* (<0.05), we performed Bonferroni-corrected pairwise comparisons. We denote these comparisons as P1 (persistent SRF vs. regressed SRF), P2 (persistent SRF vs. control), and P3 (regressed SRF vs. control) in Table 3 (or the footnotes), allowing readers to see which specific groups differ. Figure 2 illustrates the small vessel inside disc area (%) for persistent SRF, regressed SRF, and control groups, emphasizing the statistically significant difference observed between the persistent SRF and regressed SRF groups. 

Figure 3’s Top row: Enface views highlight the vascular structure in different retinal layers, including the vitreous/retina (above OPL), radial peripapillary capillaries (ILM-NFL), and choroid (below RPE).Figure 3’s Middle row: ONH analysis provides detailed measurements of cup/disc area ratio, rim area, disc area, and cup volume. Retinal nerve fiber layer (RNFL) thickness and vessel density maps are shown, illustrating structural and vascular integrity.Figure 3’s Bottom row: RNFL thickness map (left) and vessel density map (right) offer quantitative insights into the peripapillary region’s structural and vascular changes.

These findings indicate significant alterations in the optic nerve and retinal layers associated with chronic CSCR.

### 3.3. Radial Nerve Fiber Layer Thickness (RNFL)

RNFL thickness was evaluated in the peripapillary, superior, inferior, temporal, and nasal regions. Patients with persistent SRF exhibited a significant increase in RNFL thickness in the inferior and nasal quadrants compared to the healthy control group (*p* = 0.003 and *p* = 0.014, respectively). No statistically significant differences were found in the other quadrants. Table 3 shows the analysis of which groups differ in the parameters with statistical differences as a result of the comparison.

Radial peripapillary capillary density and retinal nerve fiber layer thickness differences are shown in Figure 4:

These images highlight differences in vascular density and structural integrity in the presence and absence of peripapillary involvement in CSCR patients.

### 3.4. Correlation Analyses

A significant negative correlation was found between nasal quadrant RNFL thickness and small vessel disc area in the persistent SRF group (*p* = 0.014, r = −0.306). Correlation analysis between RPC VDs and significant RNFL thickness parameters is shown in Table 4.

## 4. Discussion

This study aimed to investigate the structural changes induced by subretinal fluid (SRF) in chronic central serous chorioretinopathy (CSCR) patients, focusing on retinal nerve fiber layer (RNFL) thickness and radial peripapillary capillary (RPC) vessel density. The results demonstrate that persistent SRF, even when it does not directly involve the peripapillary region, leads to significant changes in optic nerve fibers and surrounding vascular networks. These findings align with the broader literature on CSCR but also provide new insights into the localized impact of SRF on retinal structures [12,13,14,15,16,17]. Although acute CSCR often resolves spontaneously within three to four months, the literature varies in defining chronic CSCR: some studies set a three-month threshold while others suggest six months of persistent SRF. In this study, we chose the six-month criterion to specifically target more advanced or stable cases with a reduced likelihood of spontaneous resolution. This approach aligns with previous publications indicating that persistent SRF beyond six months is often associated with more pronounced structural changes and warrants closer monitoring [18,19].

The increased RNFL thickness observed in the nasal and inferior quadrants in patients with persistent SRF aligns with existing studies but also offers unique insights. Breukink et al. documented similar thickening in the RNFL among patients with chronic CSCR, suggesting that prolonged SRF presence may lead to adaptive or compensatory structural changes within the retina to maintain its integrity. Our findings reinforce this idea, as the specific regions affected may be adapting to chronic mechanical pressure exerted by SRF. Such localized thickening in response to persistent SRF may reflect the retina’s effort to counterbalance structural stress, potentially aimed at preserving its visual function [20,21].

In our study, the observation of RNFL thickening in the nasal and inferior quadrants, rather than in regions more directly affected by macula-centered fluid, suggests a multifactorial pathophysiological process. One possibility is localized biomechanical stress, whereby even a primarily macular fluid pocket generates subtle pressure gradients affecting adjacent retinal areas. Additionally, nerve fiber topography may predispose certain fiber bundles—particularly those in the inferior retina—to compensatory swelling or reactive changes. Vascular regulation mechanisms, including regional alterations in choroidal or peripapillary blood flow, could also contribute to localized thickening. These hypotheses underscore the complex interplay between fluid distribution, nerve fiber arrangement, and microvascular factors in chronic CSCR, warranting further investigation with advanced imaging and longitudinal assessment [22,23].

Interestingly, this thickening was not uniform across all RNFL quadrants. While the nasal and inferior quadrants showed significant changes, other regions did not exhibit similar thickening. This discrepancy raises important questions about the underlying mechanisms of SRF’s impact on retinal structure, suggesting that specific retinal regions may exhibit different degrees of susceptibility or adaptive capacity to SRF-induced stress. It is conceivable that certain retinal areas have structural or cellular characteristics that make them more reactive to chronic fluid accumulation. Future studies could investigate these regional variations in greater depth, potentially utilizing advanced imaging or histological techniques to explore cellular responses in these regions [24,25]. Moreover, the baseline characteristics of chronic CSCR patients, including retinal and vascular properties, may also influence the observed differences in RNFL thickness in the nasal and inferior quadrants. While SRF plays a significant role in inducing structural changes, these baseline features might contribute to the varying susceptibility of specific retinal regions. Future studies should consider these factors to better isolate the effects of SRF and provide a more comprehensive understanding of its impact on retinal microstructure

Similarly, Ferrara et al. (2008) discussed how SRF can exert compressive effects on retinal microstructures, potentially leading to thickening in response to chronic SRF accumulation. Our findings build on Ferrara’s work by showing that these adaptive changes extend beyond the immediate area of SRF accumulation, indicating a more widespread impact on retinal structures. This suggests that even when SRF does not involve the peripapillary region, its effects can influence adjacent retinal areas, particularly in chronic cases [26].

A significant finding of our study was the decrease in RPC vessel density, particularly in the small vessel disc area. This observation aligns with Lim et al., who reported that SRF disrupts retinal venous blood flow, leading to reduced vessel density. Lim’s work focused on the general disruption of microcirculation in CSCR patients, while our study provides more specificity by highlighting the localized reduction in vessel density in areas outside the peripapillary region. The decrease in vessel density suggests that SRF not only impacts neural components of the retina but also compromises microcirculation, exacerbating damage to retinal tissues. This vascular deterioration could impair blood flow and nutrient delivery, which is crucial for maintaining the structural and functional integrity of the optic nerve fibers [27]. Several factors may contribute to this selective effect, including regional microvascular regulation, peripapillary hydrostatic pressure differences, and localized inflammatory processes. Small-caliber vessels in the disc area could be more sensitive to subtle fluctuations in tissue pressure or perfusion, even if subretinal fluid does not directly contact the peripapillary region. Further longitudinal and molecular studies would help clarify the underlying mechanisms responsible for this localized vascular response [28].

This reduction in vascular density is further corroborated by findings from Mohabati et al. (2018), who noted that severe phenotypes of chronic CSCR are associated with long-term visual decline due to both structural and vascular changes. Mohabati’s work supports our conclusion that persistent SRF contributes to vascular impairment, particularly in chronic cases. However, while Mohabati’s study focused on severe cases, our findings suggest that even in cases where SRF does not directly involve the peripapillary region, there are significant vascular consequences [29].

Our study also revealed a negative correlation between increased nasal quadrant RNFL thickness and decreased small vessel density in the optic disc area, suggesting a complex relationship between SRF and retinal microstructures. This dual impact—where SRF induces RNFL thickening while reducing vascular density—points to the multifaceted effects of SRF on the retina. Gawęcki et al. similarly found that even after SRF resolves, there are lasting changes in retinal morphology. However, unlike Gawęcki’s study, which did not isolate specific regions, our study emphasizes that structural changes can occur even in areas not directly affected by SRF. This suggests that SRF may trigger broader retinal adaptations in chronic cases, leading to structural changes that extend beyond the initial site of fluid accumulation [30,31].

Moreover, Choi et al. used ultra-high-resolution imaging to reveal cellular changes in retinal structures, which may further explain the localized thickening we observed in the RNFL [27]. Choi’s findings suggest that even subtle disruptions in the retinal microenvironment, such as those caused by SRF, can lead to profound cellular adaptations over time. This supports our hypothesis that the RNFL thickening observed in our study is an adaptive response to the chronic presence of SRF, aimed at preserving retinal function.

The findings of our study, combined with insights from previous literature, underscore the importance of early detection and management of SRF in CSCR patients. Diagnosing and managing central serous chorioretinopathy highlights that while acute CSCR often resolves spontaneously, chronic cases require more proactive management due to the risk of long-term structural damage. Our findings align with this recommendation, as they demonstrate that even SRF that does not directly involve the peripapillary region can lead to significant structural and vascular changes in the retina. This underscores the need for targeted interventions aimed at mitigating the impact of SRF on retinal and optic nerve structures to prevent long-term visual impairment [32,33].

Breukink et al. and Mohabati et al. both emphasized the long-term impact of chronic CSCR on visual function and quality of life [20,29]. While our study primarily focused on structural changes, these previous findings highlight the functional consequences of these structural alterations. The decrease in RPC vessel density observed in our study may contribute to long-term visual decline, as vascular health is critical for maintaining retinal function. This connection between structural changes and visual outcomes reinforces the need for early recognition and management of SRF in chronic CSCR patients [34,35,36].

## 5. Study Limitations and the Need for Functional Correlates

While our findings provide valuable insights into the structural consequences of SRF in chronic CSCR, several limitations must be acknowledged. First, the relatively small sample size, although sufficient for detecting significant differences, limits the generalizability of our findings. Larger, multicenter studies could provide more robust data and may uncover additional patterns that were not evident in our cohort. Furthermore, the lack of long-term follow-up restricts our ability to evaluate how these structural changes influence visual function over time. Incorporating functional visual assessments, such as visual acuity, contrast sensitivity, or quality of life measures, in future studies could help bridge the gap between structural and clinical findings in CSCR research.

Another limitation is the absence of molecular or histological analysis, which could provide a deeper understanding of the cellular and biochemical mechanisms underlying RNFL thickening and RPC vessel density reductions. Advanced imaging techniques, such as high-resolution optical coherence tomography (OCT) or histological analysis, could offer insights into cellular-level adaptations in response to chronic SRF presence. Future research integrating these methods may identify specific molecular pathways activated by SRF, potentially uncovering novel targets for pharmacological interventions to mitigate retinal damage.

## 6. Recommendations for Future Research Directions

Our study adds to the growing evidence that SRF’s impact on the retina extends beyond areas of direct involvement. However, numerous unanswered questions remain. Future research should aim to confirm these findings in larger, multicenter cohorts to enhance their applicability to broader populations. Incorporating longitudinal assessments could also help clarify how the structural changes we observed evolve over time and how they influence long-term visual outcomes. Tracking RNFL thickness and RPC vessel density at multiple time points would provide valuable insights into whether these parameters stabilize, progress, or regress with time.

Additionally, including functional assessments in future studies would be crucial for correlating structural changes with clinical symptoms. Investigating whether changes in RNFL thickness and RPC vessel density translate to visual impairments could provide a more comprehensive understanding of CSCR’s impact. This approach should involve not only conventional visual acuity tests but also advanced functional measures, such as microperimetry or multifocal electroretinography (mfERG), which can detect localized visual field defects and retinal sensitivity loss.

Finally, future studies could benefit from a more granular analysis of the biological responses to SRF in different retinal regions. Our findings suggest that certain areas, such as the small vessel disc area, may be more vulnerable to SRF’s effects, while others remain relatively unaffected. Molecular profiling of affected and unaffected regions could reveal cellular responses unique to each area, potentially guiding targeted treatments. Histological analyses combined with advanced imaging techniques could elucidate the cellular adaptations or damages caused by SRF, enabling more precise interventions to preserve retinal health.

## 7. Conclusions

This study highlights the significant structural and vascular changes in the retina of patients with chronic CSCR and persistent SRF, even when the peripapillary region is not directly involved. The findings indicate that chronic SRF leads to increased RNFL thickness and decreased RPC vessel density, particularly in areas beyond the immediate site of fluid accumulation. These results emphasize the broader impact of SRF on both neural and vascular components of the retina, contributing to a more comprehensive understanding of the pathophysiology of chronic CSCR.

Early detection and proactive management of SRF are crucial to preventing long-term structural damage and potential visual impairment in affected patients. Future research should prioritize investigating the functional consequences of these structural changes and developing optimized treatment strategies to mitigate the long-term effects of CSCR.

## Figures and Tables

**Figure 2 diagnostics-15-00174-f002:**
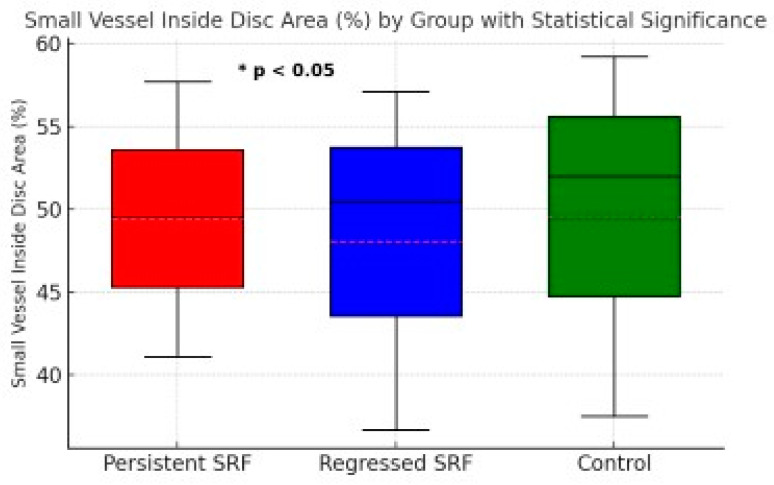
Small vessel inside disc area (%) for persistent SRF, regressed SRF, and control groups. Boxplot shows the median, interquartile range, and minimum/maximum values for each group. A statistically significant difference (* *p* < 0.05) was observed between the persistent SRF and regressed SRF groups.

**Figure 3 diagnostics-15-00174-f003:**
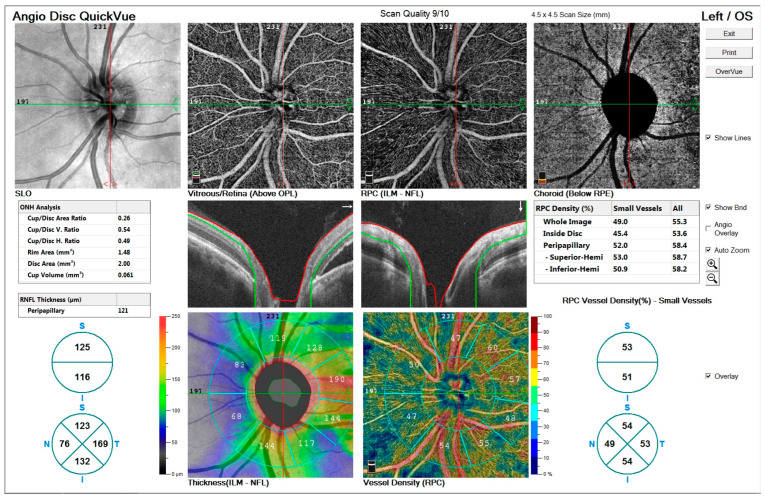
OCTA imaging of a 48-year-old male patient with peripapillary involvement due to central serous chorioretinopathy (CSCR).

**Figure 4 diagnostics-15-00174-f004:**
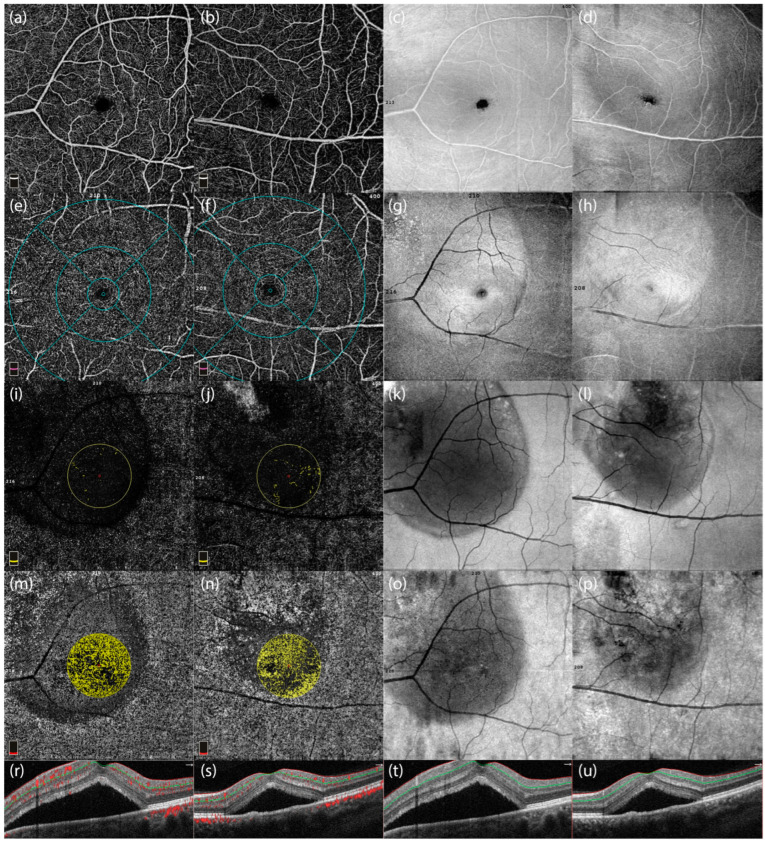
Optical Coherence Tomography Angiography (OCTA) images and corresponding optical coherence tomography (OCT) scans of two male patients with central serous chorioretinopathy (CSCR). The first and third columns (**c**,**g**,**k**,**o**) represent images from a 48-year-old male patient with peripapillary involvement, while the second and fourth columns (**d**,**h**,**l**,**p**) represent a 46-year-old male patient without peripapillary involvement. (**a**,**b**) Enface OCTA images highlighting the vascular structures of the macular region in both patients. (**e**,**f**) OCTA images showing the vascular density distribution in the central macular region, emphasizing differences between patients with and without peripapillary involvement. (**i**,**j**) Vascular perfusion maps focusing on the central macular area, showing localized reductions in vascular density in the patient with peripapillary involvement. (**m**,**n**) Color-coded maps of radial peripapillary capillary (RPC) vessel density, illustrating microvascular differences between the two cases. (**r**,**s**) Cross-sectional OCTA scans showing structural and vascular characteristics near the optic nerve head. (**t**,**u**) OCT cross-sectional scans displaying the retinal layer thickness and subretinal fluid accumulation in the two patients, highlighting the structural differences in the presence and absence of peripapillary involvement.

**Table 1 diagnostics-15-00174-t001:** Comparison of age, vision, and intraocular pressure between chronic CSCR patients with persistent and regressed SRF.

	CSCR Patients with Regressed * SRF Median (Min–Max) *n*: 27	CSCR Patients with SRF Median (Min–Max) *n*: 38	*p* Value
Age (year)	47.00 (31.00–78.00)	45.00 (32.00–78.00)	0.457
Vision (LogMar)	0.045 (0.154–0.00)	1.301 (1.00–0.096)	<0.001
IOP (mmHg)	14.00 (11.00–20.00)	13.00 (9.00–20.00)	0.469

* Regressed SRF refers to patients whose subretinal fluid has resolved, as described in the Methods section.

**Table 3 diagnostics-15-00174-t003:** Analysis of which groups differ in the parameters with statistical differences as a result of the comparison.

	*p*	P_1_ ***	P_2_ ***	P_3_ ***
RNFL Inferior Thickness (µ)	0.003	1	0.003	0.054
RNFL Nasal Thickness (µ)	0.014	1	0.023	0.053
RPC Small Vessel Inside Disc Area VDs (%)	0.021	0.538	0.019	0.940

* Bonferroni test.

**Table 4 diagnostics-15-00174-t004:** Correlation analysis between RPC VDs and significant RNFL thickness parameters.

		RNFL Inferior Thickness (µ)		RNFL Nasal Thickness (µ)	
		*p*	r ^†^	*p*	r
RPC VDs (%)	Small Vessel Whole Area	0.206	−0.160	0.048 ***	−0.249
	Small Vessel Inside Disc Area	0.393	−0.109	0.014 ***	−0.306
	Small Vessel Peripapillary Area	0.015 ***	0.300	0.434	−0.099
	All Vessel Whole Area	0.003 ****	0.358	0.498	−0.085
	All Vessel Inside Disc Area	0.431	−0.099	0.061	−0.233
	All Vessel Peripapillary Area	0.006 ****	0.335	0.835	−0.026

^†^ Spearman’s correlation test. * Statistically significant at *p* < 0.05; ** Statistically significant at *p* < 0.01.

## Data Availability

No new data were created or analyzed in this study. Data sharing is not applicable to this article.

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
