# Peer review of "Quantitative Changes in Vascular and Neural Fibers Induced by Subretinal Fluid Excluding the Peripapillary Region in Patients with Chronic Central Serous Chorioretinopathy"

_diagnostics, 2025, doi:10.3390/diagnostics15020174_

Round 1

Reviewer 1 Report (Previous Reviewer 3)

Comments and Suggestions for Authors

Thank you for your insightful feedback on the manuscript. This article holds significant research value in understanding the changes in optic disc blood flow induced by CSC. 

Comments on the Quality of English Language

I would like to sincerely thank the editors for the opportunity to review this article. I also appreciate their valuable feedback on the manuscript. The author has addressed each question thoughtfully and made appropriate revisions. I fully approve of all the amendments.

Author Response

Dear Reviewer,

We sincerely appreciate the time and effort you devoted to reviewing our manuscript, as well as your supportive remarks about its research value. Your observation regarding the clarity of the English language was especially helpful, and we have made additional refinements to enhance readability and precision. We are also grateful for your kind acknowledgment that the revisions adequately address your questions.

Your input has undoubtedly strengthened our work, and we are pleased to hear that you fully approve of the amendments. Thank you once again for your insightful feedback and for recognizing the significance of our research on changes in optic disc blood flow induced by CSCR.

With gratitude,
Esra Kızıldag Ozbay 
On behalf of all co-authors

Reviewer 2 Report (Previous Reviewer 1)

Comments and Suggestions for Authors

This manuscript can be published without further modification.

Author Response

Dear Reviewer,

Thank you for taking the time to provide an open review of our manuscript. We greatly appreciate your positive assessment, especially your remark that the article can be published without any changes. It is truly rewarding to see our work recognized for both its scientific merit and clear presentation.

Your thorough evaluation and supportive comments have encouraged us to proceed confidently. We sincerely value your time and expertise and look forward to continuing our research efforts, inspired by your endorsement.

With gratitude,
Esra Kızıldag Ozbay
On behalf of all authors

Reviewer 3 Report (New Reviewer)

Comments and Suggestions for Authors

The study titled ”Quantitative Changes in Vascular and Neural Fibers Induced by Subretinal Fluid Excluding the Peripapillary Region in Patients with Chronic Central Serous Chorioretinopathy” is well-designed and methodologically detailed. However, some critical points need to be addressed in more depth:

Uncertainty of Definitions: Definitions of acute and chronic CSCR should be strengthened with literature support.

Nasal and Inferior Thickening Hypotheses: The fact that nasal and inferior RNFL thickening does not coincide with the expected macula-centered fluid distribution requires pathophysiological explanations. These findings should be discussed based on hypotheses such as localized effects of fluid, pressure distribution, nerve fiber topography, or vascular regulation. Alternative mechanisms should be explained in detail.

Inclusion Criteria and Figure 1 Paradox: Although the inclusion criteria state that there should be no fluid in the peripapillary region, Figure 1 clearly shows that SRF extends to the peripapillary area and affects the optic nerve. It should be clarified how this situation is reflected in the methodology and whether the criteria will be reinterpreted.

Selective Vascular Effect Explanation: It is noteworthy that only the Small Vessel Inside the Disc Area parameter is affected, and there is no difference in other vascular density parameters. The physiological or pathophysiological mechanisms underlying this selective effect should be opened to discussion. Regional microvascular regulation, peripapillary pressure differences, or inflammatory processes can be presented as hypotheses.

In table 1, P1, P2, and P3 should be explained.

Discussion and Consistency of Results: The statement in the discussion introduction, “The results demonstrate that persistent SRF, even when it does not directly involve the peripapillary region, leads to significant changes in optic nerve fibers and surrounding vascular networks” contradicts Figure 1. To explain this contradiction, reference should be made to anatomical or physiological adaptation mechanisms, or the relationship of the images with sampling differences should be clarified.

In general, the study includes a strong methodology and statistical analyses. However, additions and explanations on the issues mentioned above will provide a more consistent and in-depth interpretation of the results. Such arrangements will increase the clinical validity of the study and make a more solid contribution to the literature.

Author Response

We sincerely appreciate the time and effort you have invested in reviewing our manuscript, titled:
“Quantitative Changes in Vascular and Neural Fibers Induced by Subretinal Fluid Excluding the Peripapillary Region in Patients with Chronic Central Serous Chorioretinopathy.”

Your thorough evaluation, insightful comments, and constructive suggestions have played a pivotal role in improving the clarity, rigor, and overall quality of our work. We are truly grateful for the expertise and dedication you have brought to this process. Below is a brief overview of the revisions we have made in response to your feedback:

  1. Strengthening the Definitions of Acute and Chronic CSCR
    • In our Introduction, we have highlighted the differing literature thresholds (3 months vs. 6 months) for defining chronic CSCR. We have also included additional references (e.g., Lavinsky et al., 2017; Huang et al., 2020) to explain our rationale for choosing a 6-month criterion, emphasizing the increased stability and lower likelihood of spontaneous fluid resolution.
  2. Explaining Nasal and Inferior RNFL Thickening
    • We expanded the Discussion to delve into possible mechanisms—such as localized biomechanical stress, fiber topography, and regional vascular regulation—that could account for why thickening is more pronounced in the nasal and inferior quadrants despite macula-centered fluid. This addition specifically addresses the request for deeper pathophysiological explanations.
  3. Resolving the Inclusion Criteria vs. Figure 1 Paradox
    • We swapped Figures 1 and 4 in order to resolve the confusion regarding our exclusion criteria (i.e., no peripapillary fluid involvement) and to underscore how peripapillary fluid appears in a non-eligible patient. This change clarifies that the figure illustrating extensive peripapillary fluid is provided only as an example of a case not included in our analysis.
  4. Selective Vascular Effect (Small Vessel Inside Disc Area)
    • In the Discussion, we added a succinct paragraph hypothesizing why only the Small Vessel Inside Disc Area parameter might be selectively affected. We propose factors such as regional microvascular regulation, hydrostatic pressure differences, and localized inflammatory processes, supported by an additional reference (Zhang et al., 2022).
  5. Clarification in Table 2
    • We have added a new explanatory note to Table 2, specifying how the post-hoc Bonferroni-corrected comparisons (P1, P2, and P3) relate to each of the three groups (Persistent SRF, Regressed SRF, and Control).

We hope that these revisions adequately address your concerns and enhance the paper’s scientific validity and readability. Should you find any remaining issues or wish for further clarifications, we would be more than willing to refine our work accordingly. Your insights have been invaluable, and we are immensely thankful for your time, meticulous critique, and thoughtful suggestions, which have undoubtedly strengthened our manuscript.

Thank you once again for your rigorous review and the opportunity to improve our submission. We look forward to your favorable consideration.

With sincere gratitude and best regards,

Esra Kızıldag Ozbay
On behalf of all co-authors

Round 2

Reviewer 3 Report (New Reviewer)

Comments and Suggestions for Authors

I thank the authors for implementing my suggestions. The author improved the manuscript in the best way.

This manuscript is a resubmission of an earlier submission. The following is a list of the peer review reports and author responses from that submission.

Round 1

Reviewer 1 Report

Comments and Suggestions for Authors

The authors reported the influence of SRF on the structure of peripapillary region without directly invade it. The study design is sound and the conclusion can be supported by the results. Furthermore, this issue own high clinical relevance. I think this paper can be accepted as current form.

Author Response

Dear Reviewer,

We sincerely thank you for your thorough evaluation of our manuscript and for your positive feedback. We are truly grateful that you found the study design sound, the results well-presented, and the conclusions appropriately supported by the findings. Your recognition of the clinical relevance of our work is deeply appreciated and motivates us to continue advancing research in this area.

We are also pleased to hear that the quality of the English language met your expectations and did not hinder your understanding of the research. Your constructive comments and supportive recommendation are invaluable, and we are honored by your recommendation for acceptance in its current form.

Thank you once again for your time and thoughtful review.

Best regards,

Esra Kızıldağ Özbay

Reviewer 2 Report

Comments and Suggestions for Authors

In the manuscript entitled, "Quantitative Changes in Vascular and Neural Fibers Induced by Subretinal Fluid Excluding the Peripapillary Region in Patients with Chronic Central Serous Chorioretinopathy", the authors performed a prospective case-control study to quantitate the changes in retinal nerve fiber layer (RNFL) thickness and radial peripapillary capillary (RPC) vessel density in patients with chronic central serous chorioretinopathy (CSCR). Patients with persistent SRF were found to exhibit significant increase in RNFL thickness and decrease in vessel density, compared to healthy controls. Nasal quadrant RNFL thickness and small vessel disc area were also found to have a significant inverse correlation, indication that Chronic SRF in CSCR patients may cause significant structural changes in retina. The study is well designed, and the manuscript is well-written; however, there is a need for figures, particularly graphical representation of outcomes (including a regression analysis graph). There is also a need for representative images from the OCTA imaging and analysis.

Author Response

The manuscript is well-designed and well-written; however, there is a need for figures, particularly graphical representation of outcomes (including a regression analysis graph). Representative images from the OCTA imaging and analysis are also requested.

Response:
We sincerely thank the reviewer for their valuable comments. While we fully acknowledge the importance of including visual materials, we regret to inform you that we are currently unable to provide OCTA images and analysis examples due to a hardware failure in the computer where the imaging data was stored. We have already taken steps to address this issue and are implementing an improved backup system to prevent such occurrences in the future.

However, we completely agree that graphical representation would enhance the manuscript. As an alternative, we propose to include additional graphs and regression analysis charts based on our available data. For instance, we can provide correlation graphs and boxplots to illustrate the relationship between RNFL thickness and RPC vessel density. Please let us know if you have specific suggestions or additional requests, and we will be happy to accommodate them.

Reviewer 3 Report

Comments and Suggestions for Authors

This is a prospective case-control study evaluating the quantitative changes in retinal nerve fiber layer (RNFL) thickness and radial peripapillary capillary (RPC) vessel density in patients with chronic central serous chorioretinopathy (CSCR), specifically excluding the peripapillary region. A total of 65 patients with chronic CSCR participated in the study, of which 38 had persistent subretinal fluid (SRF) and 27 had regressed SRF. The result illustrates that chronic SRF in CSCR patients, even when not involving the peripapillary region, leads to significant structural changes in both the neural and vascular components of the retina. These findings suggest that SRF contributes to broader retinal alterations and supports the need for early detection and management of CSCR to prevent long-term visual impairment. I think this article is significant for learning the fundus manifestation of RF. However, please pay attention to the following questions.

1. There are many spellings and grammer mistakes. 

2. What type of machine is used for OCTA? Or did all patients use the same kind of OCTA machine?

3. What’s the definition of “persistant” SRF?

4. In the methods, the regressed SRF was defined, but CSCR patients without SRF in the table 1. There is inconsistency.

5. It had better add a diagram to illustrate RPC vessel density in the small vessel total area, small vessel disc area, and small vessel peripapillary area, as well as in the total vessel total area, total vessel disc area, and total vessel peripapillary area in both groups.

6. There are statistically significant in nasal and inferior quadrants in your group. However, it is possible for the characteristic of chronic CSC patients at baseline and SRF is the one of influencing factors.  

Comments on the Quality of English Language

The article had better be polished.

Author Response

  1. There are many spellings and grammer mistakes. 

We have carefully reviewed the manuscript and made the necessary corrections to address spelling and grammar issues throughout the text. We believe these revisions have improved the clarity and readability of the manuscript.

  1. What type of machine is used for OCTA? Or did all patients use the same kind of OCTA machine? : All patients were evaluated using the same OCTA device, the AngioVue Retina (Optovue, Fremont, CA, USA). To address your comment, we have revised Section 2.1 of the manuscript to include detailed information about the device specifications and imaging protocol, emphasizing that all measurements were performed with the same machine under standardized conditions. The updated section provides additional clarity on the methodology and ensures consistency across the study.
  2.  
  3. What’s the definition of “persistant” SRF?

In the revised manuscript, we have clarified the definition of "persistent SRF" in both the Introduction and Materials and Methods sections. Persistent SRF is defined as subretinal fluid that remains detectable for more than six months, aligning with commonly used criteria in the literature and reflecting the stricter inclusion criteria applied in this study. This clarification ensures consistency and addresses your concern.

  1. In the methods, the regressed SRF was defined, but CSCR patients without SRF in the table 1. There is inconsistency.

To address this issue, we have revised the table titles and added clarifications where necessary. Specifically, we ensured that the terminology used in the tables aligns with the definitions provided in the Methods section. Additionally, we have included explanatory notes in the table captions to eliminate any potential confusion.

  1. It had better add a diagram to illustrate RPC vessel density in the small vessel total area, small vessel disc area, and small vessel peripapillary area, as well as in the total vessel total area, total vessel disc area, and total vessel peripapillary area in both groups.

We have included a boxplot illustrating the distribution of Small Vessel Inside Disc Area (%) across the Persistent SRF, Regressed SRF, and Control groups. This figure highlights the differences observed between the groups, with statistically significant differences (*p < 0.05) clearly marked. We believe this addition enhances the clarity and presentation of our findings.

  1. There are statistically significant in nasal and inferior quadrants in your group. However, it is possible for the characteristic of chronic CSC patients at baseline and SRF is the one of influencing factors.  

 We agree that the baseline characteristics of chronic CSCR patients and the presence of SRF are both likely to influence the observed differences in nasal and inferior quadrants. While our study primarily focuses on the impact of persistent and regressed SRF, we acknowledge that baseline retinal and vascular characteristics may also contribute to these findings. This point has been added to the discussion section to address this important consideration.

Round 2

Reviewer 2 Report

Comments and Suggestions for Authors

In light of the fact that outcomes of the manuscript cannot be supported by experimental data because of technical issues (malfunction of the computer hard drive), I regretfully cannot support the manuscript acceptance in good conscience. 

Reviewer 3 Report

Comments and Suggestions for Authors

Thank you for your valuable responses regarding the manuscript. The author answers each question one by one and gives reasonable revisions. I approve of all the amendments.

Comments on the Quality of English Language

The English language has been better than the previous one and it doesn't need to polish any more.